# Study on the Properties and Heavy Metal Solidification Characteristics of Sintered Ceramsites Composed of Magnesite Tailings, Sewage Sludge, and Coal Gangue

**DOI:** 10.3390/ijerph191711128

**Published:** 2022-09-05

**Authors:** Yanlong Li, Mingyuan Xu, Quan Li, Anjun Gai, Tianhua Yang, Rundong Li

**Affiliations:** The Key Laboratory of Clean Energy in Liaoning Province, College of Energy and Environment, Shenyang Aerospace University, Shenyang 110136, China

**Keywords:** magnesite tailings, sewage sludge, ceramsite, heavy metals, OPTI evaluation

## Abstract

With the rapid development of industry, the disposal of industrial solid waste needs to be solved urgently in China. Thus, an effective disposal method should be proposed to recycle these solid wastes in an environmentally friendly and sustainable manner. In this paper, ceramsite was prepared from sewage sludge (SS), magnesite tailings (MTs), and coal gangue (CG). The influence of the material ratio and sintering temperature on the properties of the ceramsite was investigated. The results show that the ceramsite had better properties when the following parameters were used: a ratio of SS: CG: MT of 4.5:4:1.5; a sintering temperature of 1250 °C; a compressive strength of 11.2 MPa (or it can be rounded to 11; our major remark relates to significant figures, and they should be up to 2–3 figures, according to measurement errors); a water absorption of 3.54%; and apparent and bulk densities of 1.19 and 0.81 g/cm^3^, respectively. The strength was superior to more than twice the 900-density grade prescribed by the Chinese national standard. After sintering, most of the heavy metals in the ceramsite mainly existed in the form of residue state (FD), meaning that they were highly stable. The leaching concentrations of Zn and Ni from the ceramsite were 0.72 and 0.25 mg/L lower than the prescribed regulatory limits (2.0 and 0.1 mg/L). The overall pollution toxicity index (OPTI) was only 240, less than that of raw pellets, indicating that the environmental risk is low. Not only did the ceramsite, prepared from SS, CG, and MT, exhibit excellent chemical properties, but it also proved to be an environmentally safe material. Therefore, it is an effective approach to realize the collaborative treatment of SS, CG, and MT by preparing ceramsite.

## 1. Introduction

With the rapid development of China’s industry, the disposal of industrial solid waste needs to be solved urgently [1]. China has large reserves and a high mining volume of magnesite, and the proved reserves are about 3.6 billion tons, accounting for 30% of the world reserves, and the annual mining volume reaches 20 million tons. If the tailings generated by mining are not properly treated, they will pollute the environment and endanger human life [2]. Calcination and sintering are currently the most mature tailing disposal methods. The preparation methods of MgCO_3_ and MgO comprise calcination, hydration, carbonization and thermal decomposition methods, and the preparation method of magnesiaalumina spinel cementitious material is sintering [3]. However, the disadvantages of high energy consumption and incomplete mineral separation are difficult to avoid. In addition, the disposal of SS and CG has also become a difficult problem that cannot be ignored. SS production reached 70 million tons in China in 2020 [4,5], and the annual accumulation of CG also reached 280 million tons [6,7,8]. Incineration is the most mature treatment method of SS, and the treatment of coal gangue is mainly for power generation and road paving. However, the existing methods have the problems of causing secondary pollution and having a low comprehensive utilization rate. Thus, an effective disposal method should be proposed to recycle these solid wastes in an environmentally friendly and sustainable manner [9].

Ceramsite is widely used in building materials due to its excellent properties such as its low density, low water absorption, and high strength. Its further development to replace traditional building materials is economically beneficial. Recently, the Chinese government introduced policies to restrict the mining of conventional ceramsite raw materials, such as shale and clay [10]. Thus, it is an urgent issue to find new materials to replace the original ceramsite raw materials in order to achieve the purpose of sustainable development. The synergistic preparation of ceramsite using a variety of industrial solid wastes, such as sludge, coal ash, waste rock, and tailings, can take full advantage of the valuable components of industrial solid wastes and avoid secondary pollution. Riley [11] identified two conditions necessary for the sintering of ceramsite. First, sufficient gas can be generated during the sintering process; the second is that the liquid phase can be generated during the sintering process, and the liquid phase has enough viscosity to effectively wrap the gas generated inside the green body. Nakouzi et al. [12] first prepared ceramsite by using sludge in 1998, and the ceramsite formed a hard surface during high-temperature sintering. Cheng et al. [13] used sludge, fly ash and oyster shell to sinter ceramsite, and they obtained an optimal phosphate adsorption capacity of 4.51/mg at a sintering temperature of 1050 °C. Bouachera et al. [14] mixed sewage sludge, waste glass, and clay at a certain quality ratio (3:3:4) to reduce the sintering temperature and to lower the bulk density of ceramsite to 0.84 g/cm^3^, but the compressive strength was only 2.51 MPa. Liu et al. [15] used sewage sludge and river sediment to produce ceramsite, and they found that ceramsite with optimum engineering properties can be prepared when the SiO_2_ and the Al_2_O_3_ contents are 30–45% and 11–19%, respectively. These studies provide ideas for the preparation of ceramsite from industrial solid waste, but their disadvantage is that they could not improve the strength of ceramsite while maintaining the excellent environmental safety characteristics of ceramsite.

At present, there are few studies on the preparation of ceramsite with magnesite tailings as raw materials. Magnesite tailings contain sufficient MgO; the Mg^2+^ contained in MgO can cause the transformation of the crystalline phase in the ceramsite, which improves the strength of the ceramsite to a certain extent. Furthermore, Mg^2+^ can be incorporated into the SiO_2_-Al_2_O_3_-CaO ternary vitreous phase to change its viscosity and produce a denser surface layer, thereby decreasing water absorption. In this paper, sewage sludge (SS), magnesite tailings (MTs), and coal gangue (CG) were used as raw materials to prepare ceramsite. By adjusting the ratio of the raw materials and changing the sintering temperature to prepare ceramsite with favorable properties, the material properties were analyzed. Next, inductively coupled plasma-optical emission spectrometry (ICP-OES), BCR sequential extraction [16], and the toxicity characteristic leaching procedure (TCLP) [17] were used to detect and analyze the concentration, the leaching concentration, and the occurrence state of the heavy metals in the ceramsite, and the OPTI [18] index was combined to evaluate the environmentally safe characteristics of them. The purpose of this study was to determine the feasibility of using MTs, SS, and CG to prepare environmentally friendly artificial ceramsite with excellent performance; this provides a solution for the resource utilization of industrial solid waste.

## 2. Materials and Methods

### 2.1. Characterization of Materials

The MTs, SS, and CG used in this study were sampled from Anshan, Shenyang, and Lingshou, respectively. The moisture content of SS was high (about 82%). First, these three materials were dried for 48 h in an oven and then ground as well as passed through 100 mesh sieves (≤0.154 mm). Table 1 and Table 2 show the chemical compositions determined by conducting X-ray fluorescence (PANalytical Axios, XRF, Eindhoven, the Netherlands) and an elemental analysis (Elementar Vario EL, Langenselbold, Germany) of these materials. It can be seen that the three raw materials contain SiO_2_. The main component of the magnesite tailings is MgO; this exists in the form of MgCO_3_ and contains a large amount of volatile matter, which produces gas during the process of sintering. The levels in the SS are high. It also contains a certain amount of Fe_2_O_3_ and P_2_O_5_ [19]. The SiO_2_ and Al_2_O_3_ contents in CG are high, and it has almost no other components. SiO_2_ and Al_2_O_3_ are the main components of ceramsite’s skeleton.

### 2.2. Methods

#### 2.2.1. Experimental Operation Process

First, the SS, MTs, and CG were mixed at different ratios (Table 3 shows the mixing ratios of SS, MTs, and CG), and then the mixture was stirred and dried for 6 h in an oven at 105 °C. Furthermore, the mixture was pressed into raw pellets using a tablet press (HY-12, Changzhou, China).

The sintering experiment was accomplished using a tube furnace (Zhonghuan, SK-G08163, Tianjin, China). The raw pellets were put on a corundum dish, and the dish was placed in the center of the tube furnace. An air compressor (Daertuo, XDT550-30L, Zhejiang, China) was used to input air at a flow velocity of 150 mL/min, and the gas flow was controlled by a flow meter (Qixinghuachuang, D07-19B, Beijing, China). The experimental device is shown in Figure 1. The sintering process was controlled by the sintering program of the tube furnace, and the corundum dish was removed after the program was over. After standing at room temperature, sintered ceramsite products were obtained. The exhaust gas generated during the experiment was discharged from the rear of the furnace and passed into the dilute hydrochloric acid solution for scrubbing. In order to further study the influence of different sintering temperatures on the properties of the ceramsite, the experiment was carried out with a raw material ratio of 4.5:4:1.5.

#### 2.2.2. Characterization of Ceramsite

Thermogravimetry and a differential scanning calorimeter (TG-DSC, STA 449 F3 Jupiter) were adopted to analyze the thermal reaction process of the raw materials. X-ray diffraction (Daojin, XRD-7000S, Kyoto, Japan) was selected to characterize the crystal phase of the ceramsite. The data at different sintering temperatures were collected at an angle range of 15–80° 2θ and a scanning electron microscope (SEM, FEI, Nova Nano SEM 450, Waltham, MA, USA) was used to analyze the micro-morphology of the ceramsite.

The concentrations of the heavy metals were measured using an inductively coupled plasma optical emission spectrometer (ICP-OES, Optima PE8300, Waltham, MA, USA), and the morphological occurrence and leaching characteristics of the heavy metals were analyzed using the BCR sequential extraction procedure (the fractionation extraction method proposed by the European Community Bureau of Reference) [20] and the toxicity characteristic leaching procedure (TCLP) [21,22,23].

The overall pollution toxicity index (OPTI) was used to evaluate the environmental safety of the ceramsites. The details of the OPTI are shown in our previous study [18] (“A new integrated evaluation method of heavy metals pollution control during melting and sintering of MSWI fly ash”).

In this paper, the following four parameters were used to characterize the properties of the ceramsite. The compressive strength of the ceramsite was calculated using Equation (1):(1)Fa=FS
where *F* is the force (KN), s is the cross-sectional area under pressure (mm^2^), and *Fa* is the compressive strength (MPa).

The water absorption was calculated using Equation (2):(2)w=m1−m0m0×100%
where *m*_0_ is the quality of the ceramsite (g), *m*_1_ is the water content of the submerged ceramsite (g), and w is the water absorption rate (%).

The apparent density was calculated using Equation (3):(3)Θ=MVt
where *M* is the weight of the ceramsite (g), *V_t_* is the volume of the ceramsite (cm^3^), and Θ is the apparent density (g/cm^3^).

The bulk density was calculated using Equation (4):(4)Θbd=Mb2−Mb1Vb
where *M_b_*_1_ and *V_b_* are the weight (g) and the volume (mL) of the measuring cylinder, respectively; *M_b_*_2_ is the weight of the measuring cylinder filled with ceramsite (g); and $\Theta_{\mathrm{bd}}$ is the bulk density (g/cm^3^).

## 3. Results and Discussion

### 3.1. Thermal Characteristics of Materials

From the TG-DSC curve of the magnesite tailings (Figure 2a), it can be observed that the weightlessness is 47% between 580 °C and 700 °C, and the maximum weight loss rate is 0.59%/min at 680 °C. This is mainly due to the thermal decomposition of MgCO_3_ [24,25]. The maximum heat flux at 685 °C is also confirmed by the differential scanning calorimeter curve to be 2.35 mW/mg. The weightlessness of SS is about 52% during the whole process, and it can be divided into two stages (Figure 2b). The first stage of weightlessness is about 18%, the start time of weightlessness is 240 min, the end time of weightlessness is 300 min, the maximum weightlessness rate is 6.35%/°C, and the weightlessness time is 297 min. This is because the C-C bond of the organic matter in the SS ruptures during the reaction [26]. The weightlessness is about 28% in the range of 300–500 °C, the maximum weight loss rate is 3.1%/min at 417 °C, and the SS shows a high degree of the exothermic effect. This is due to the oxidizing and decomposing of organic macromolecules and some minerals [27]. It can be seen from the DSC curve that the maximum heat flow rate of 17.7 mW/mg occurs at 490 °C. The decomposition of carbonates, such as CaCO_3_ and MgCO_3_, produces a certain weightlessness at 840 °C. At 1000 °C, Fe_2_O_3_ reacts with the carbon from the SS pyrolysis to generate CO, and FeS_2_ oxidizes to generate SO_2_, resulting in a small amount of weight loss due to gas evolution. The weightlessness of the coal gangue mainly occurs at 470–700 °C (Figure 2c), and the weightlessness rate reaches up to 2.3%/min at 541 °C. This is due to the kaolinite undergoing the dehydration reaction and forming metakaolinite, as well as the combustion of carbon and volatile matter [28]. In addition, the combustion of carbon and volatile matter releases heat. In the temperature range of 1000–1050 °C, there is obvious heat flow fluctuation; this may be due to the transformation of the metakaolinite phase into the amorphous Al-Si spinel phase. Al-Si spinel transforms into mullite crystals at 1050 °C [29,30].

### 3.2. The Properties of Sintered Ceramsite

#### 3.2.1. The Properties of Sintered Ceramsite with Different Material Ratios

In order to determine the best mixing ratio, the ratios in Table 3 were used for experiments. The preheating temperature, preheating time, sintering temperature, and sintering time were set at 350 °C, 15 min, 1225 °C, and 20 min, respectively. The material properties of the sintered products were tested according to the Chinese national standard GB/T 17431.2–2010 [31].

As shown in Figure 3, the compressive strength of the ceramsite ranges from 3.62 MPa to 7.85 MPa; it increases first and then decreases as the content of SS decreases. When the SS content is higher than 50%, the SiO_2_ content exceeds 45%, and a more liquid phase is produced, which increases the melting degree of the ceramsite at high temperatures and decreases the strength of the ceramsite [32,33]. The compressive strength of the ceramsite (SS:CG:MT = 4.5:4:1.5) reaches 7.85 MPa, higher than that of the others, and the compressive strength is superior to the 900 density grade (5 Mp) in the Chinese national standard GB/T 17431.1–2010 [34]. In addition, the water absorption rate is 2.89%, and the apparent density and bulk density are 1.21 g/cm^3^ and 0.83 g/cm^3^ at this ratio, respectively. These properties also meet the requirements in the standard. The compressive strength of the ceramsite decreases as the proportion of magnesite tailings increases, and this could be attributed to Mg^2+^, which affects the formation of the silicate framework [35,36]. When the content of magnesite tailings is 15% and that of sewage sludge is 45%, a certain amount of molten Mg^2+^ ions penetrates the ceramic matrix and forms high-strength crystallites, such as cordierite (Mg_2_Al_4_Si_5_O_18_), which contributes to the generation of high-strength ceramsite. However, as the MgO content continues to increase, excess Mg^2+^ ions leads to the formation of a low-viscousity glass phase, breaks the structure of the ceramsite, and decreases the strength of the ceramsite. With magnesite tailing contents from 10% to 20%, the compressive strength is in the range of 5.64 MPa–7.85 MPa, higher than that of general ceramsites and meets the standard.

The water absorption increased when the content of magnesite tailings increased. Excessive MgO caused the release of a large amount of gas, and this caused microcracks and pores in the ceramsite surface, which increased the water absorption. When the ratio of SS:CG:MT was changed from 5.5:4:0.5 to 3.5:4:2.5, the morphology of the ceramsite changed from a highly molten globule to an ellipsoid with a lower expansion. The density of the ceramsite decreased as the ratio was changed. By taking into account the requirements of compressive strength in engineering applications and by combining the ceramsite for low water absorption and low density, the optimum ratio was SS:CG:MT = 4.5:4:1.5. This ratio was used, and the preheating temperature, preheating time, and sintering time were maintained to carry out subsequent experiments.

The effects of the sintering temperatures on the properties of the ceramsites are shown in Figure 4. In Figure 4, we can see that the sintering temperature has a major influence on the compressive strength and water absorption of the ceramsites. A sintering temperature that is too high or too low forms a liquid phase with a regular ratio. The compressive strength increased from 4.45 MPa to 11.21 MPa when the sintering temperatures were increased from 1220 °C to 1250 °C and decreased from 1250 °C to 1270 °C. This is due to the incomplete reaction of the raw materials at 1220 °C, insufficient liquid phase formation, and low surface enamelization, resulting in a low compressive strength [37]. At a sintering temperature of 1250 °C, the amount of liquid phase increased; furthermore, at this surface tension and viscosity of the liquid phase, the gas pressure inside the ceramsite reached a dynamic balance [38]. On the one hand, the liquid phase sintering reaction can be better promoted; on the other hand, the liquid phase generated by the sintering reaction can more easily fill in the pores. This makes the ceramsite densify and forms a uniform pore structure. Thus, the ceramsite reached the highest compressive strength of 11.21 MPa, and the water absorption, apparent density, and bulk density were 3.54%,1.19 g/cm^3^ and 0.81 g/cm^3^, respectively. The compressive strength is superior by more than twice the 900-density grade in the Chinese national standard GB/T 17431.1-2010, and the other properties also meet the requirements. At 1270 °C, due to oversintering, part of the glass phase was dissolved and formed a large number of liquid phases. The ceramsite structure was damaged, and the compressive strength was affected.

The water absorption decreased from 5.32% to 2.74% with an increase in the temperature because increasing the sintering temperature makes the surface of the ceramsite become rougher. Taken as a whole, the apparent density and bulk density increased with an increase in the sintering temperature; at 1270 °C, the minimum apparent density and bulk density were 1.17 g/cm^3^ and 0.79 g/cm^3^. This was attributed to the increase in the temperature, which increases the internal porosity of the ceramsite. Therefore, to obtain ceramsite with excellent properties, the optimum sintering temperature was determined to be 1250 °C.

#### 3.2.2. Crystal Phase Analyses

XRD was used to analyze the crystal phases of the raw materials and ceramsites. Figure 5a shows the mineral phases of the MTs, SS, and CG. It can be seen that the main mineral phase of the magnesite tailings is MgCO_3_ and a small amount of CaCO_3_, and the main mineral phases in SS and CG are quartz (SiO_2_) and kaolinite (Al_4_[Si_4_O_10_](OH)_8_).

It can be seen in Figure 5b that cordierite (Mg_2_Al_4_Si_5_O_18_), magnesia–alumina spinel (MgAl_2_O_4_), and magnesium silicate (MgSiO_3_) are the main mineral phases of ceramsites. We can conclude that a series of complex glass crystal reactions occurred in the sintering process. The main crystal phase cordierite in ceramsite is the reaction of MgO decomposed from the MgCO_3_ in the MTs with Al_2_O_3_ in SS and CG to form magnesia–alumina spinel (MgAl_2_O_4_); then, the magnesium-aluminum spinel reacts with quartz (SiO_2_) to form cordierite. Before 1250 °C, the intensity of the cordierite diffraction peak is relatively high. This stage is the generation and development stage of cordierite, which guarantees the strength of the ceramsite [39]. Above 1250 °C, the strength of the orthogonal peaks of the magnesia-alumina spinel increases. This is because the cordierite begins to decompose into magnesia-alumina spinel and amorphous silica, which affects the strength of the ceramsite.

#### 3.2.3. Morphology Structure Analyses

The micromorphology of the ceramsites was analyzed using SEM. Figure 6 shows the surface microstructures of the ceramsites at different sintering temperatures from 1220 °C to 1270 °C. At 1220 °C, the surface enamel degree was low, the complete glass phase was not formed, and there was no obvious pore structure, which caused a weak compressive strength. When the temperature was increased to 1250 °C, the outer surface of the ceramsite became dense, a more complete glass phase formed, and the compressive strength was the highest at this time. The dense surface reduced the water absorption of the ceramsite. When the sintering temperature was increased to 1270 °C, due to excessive sintering, obvious cracks appeared on the surface of the ceramsite, which resulted in a decrease in the compressive strength.

Figure 7 shows the interior microstructures of the ceramsites sintered at different temperatures; changes in the pores inside the ceramsites can be observed at different sintering temperatures. When the sintering temperature was 1220 °C, the liquid phase generated internally was not complete, the gas production was insufficient, and more pores with smaller diameters formed. When the temperature was increased to 1250 °C, the sintering effect of the ceramsite was better, and a structure with many micro pores distributed around a small number of large pores formed inside the ceramsite, which could increase the compressive strength of the ceramsite. At 1270 °C, due to oversintering, the ceramsite produced too much liquid phase; the liquid phase wrapped the gas so that it could not escape [40]. The gas gathered and expanded to thin the wall of the pores and increased the pore size, resulting in a decrease in the compressive strength; this also reduced the density of the ceramsite. In addition, the porous structure also contributed to improve the heavy metal solidification effect of the ceramsite.

### 3.3. The Solidification Characteristics of Heavy Metals at Different Sintering Temperatures

#### 3.3.1. Total Heavy Metals Concentration in Ceramsite

Figure 8 shows the heavy metal concentrations in the ceramsites under different sintering temperatures and before sintering. The contents of Zn, Mn, Cu, Cr, Ni, and Pb in the raw pellets were 1151.52, 579.51, 361.5, 270.48, 258, and 77 mg/kg, respectively. When the sintering temperature was increased from 1220 °C to 1270 °C, the heavy metal concentrations in the ceramsites decreased significantly. The heavy metals Zn, Mn, and Ni, with high contents in the raw materials, were significantly decreased At 1250 °C, the concentrations of Zn, Mn, and Ni in the ceramsites decreased to 336, 222, and 125 mg/kg, respectively.

This is because, when the sintering temperature is higher than 1200 °C, the boiling point of the heavy metal compound is reached, the heavy metal volatile matter in the flue gas [41] is absorbed by the scrubber, and the ceramsite heavy metal concentration undergoes a significant reduction.

#### 3.3.2. Morphological Occurrence of Heavy Metals in Ceramsite

Figure 9 shows the occurrence of heavy metals in the ceramsites at different sintering temperatures. Heavy metals exist in ceramsites in four states, namely, the weak acid state (FA), the reducible state (FB), the oxidizable state (FC), and the residue state (FD). The proportions of FA, FB, and FC are relatively high in unsintered raw pellets. FA and FB have direct toxicity because of their high bioavailability, and FC has potential toxicity, meaning that toxicity can easily be released under specific conditions. After high-temperature sintering, more than 95% of heavy metals exist in the form of residue (FD), and the proportions of the other forms are significantly reduced. FD is stable and does not produce toxicity.

Figure 9 shows that the proportions of Zn, Ni, and Mn in the form of FA were as high as 10.29%, 7.19%, and 8.13%, respectively, when the pellets were not sintered. In addition, the forms of FB also reached 6.9%, 6.3%, and 5.7%. These heavy metal ions exist in chlorides or carbonates under the weak acid state, and they easily break into free particles under natural conditions, and they are considered an environmental risk [42]. The FC of Cu and Cr formed at high proportions of 14.26% and 4.35%, respectively; their potential toxicities can be released under special climatic and geographical conditions, and they are also not be ignored. Pb mainly exists in the form of FD. This is due to it being difficult to release Pb from minerals under natural conditions with less mobility. However, after high-temperature sintering above 1220 °C, all the heavy metals in the ceramsites mainly exist in the form of FD. At a 1250 °C sintering temperature, Zn, Ni, and Mn ions in the form of FA and FB decreased to 0.99%, 0.62%, and 2.68%, and 0.91%, 1.29%, and 0.35%, respectively, and Cu and Cr in the form of FC also greatly reduced. Most heavy metals in FD exist as mineral components. High-temperature sintering promotes the fixation of heavy metals and enhances stability.

#### 3.3.3. Heavy Metal Leaching Characteristics of Ceramsite

Table 4 shows the results of the heavy metal leaching test of the ceramsite. The leaching concentrations of Zn, Ni, and Cu in the raw pellets were as high as 7.034, 1.079, and 5.683 mg/L, respectively. This shows that the heavy metals in the unsintered pellets leached at concentrations that were higher than the specified value. The concentrations of the leached metals decreased as the sintering temperature was increased [43]. At 1250 °C, the concentrations of leached Zn, Ni, Cu, and Cr were 0.722, 0.249, 0.887, and 0.201 mg/L, respectively, all below the limit. Pb was undetected. The high-temperature sintering of ceramsite can effectively reduce leaching concentrations.

High-temperature sintering can effectively reduce the leaching concentrations of the heavy metals in ceramsites. As the temperature increases, microstructure changes and phase changes occur in ceramsites. The microstructure of ceramsites is mainly made up of the glass phase and pores, and they are connected by a solid-state reaction bridge. Most heavy metals exist in the form of ions in the glass phase after sintering and are effectively solidified. The composite thermal reaction occurs between the solid phases by means of the solid-state reaction bridge, and free particles transform into a more stable crystal structure. The mineral composition of ceramsites comprises cordierite, magnesium-aluminum spinel, and magnesium silicate. These aluminosilicates and magnesium silicates can adsorb heavy metal ions. Heavy metal ions with smaller particle sizes migrate into aluminosilicate and magnesium silicate crystal lattices at a high temperature, so heavy metals are fixed in the mineral for a long period of time. This also means that preparing environmentally friendly ceramsites using high-temperature sintering is a meaningful method to make use of MTs, SS, and CG.

#### 3.3.4. The Comprehensive Toxicity Evaluation Index of Ceramsite

Figure 10 shows the OPTI index of the ceramsites with different sintering temperatures. Before sintering, the OPTI index of the raw pellet was high, even reaching 1506.49, which is harmful to the environment. When the sintering temperature was increased to 1250 °C, the OPTI index of the ceramsites was only 236.92, indicating that the ceramsites are generally safe.

## 4. Conclusions

(1)Under the conditions of an SS:CG:MT ratio and a sintering temperature of 4.5:4:1.5 and 1250 °C, respectively, the compressive strength of the ceramsite reached 11.21 MPa, superior to the 900-density grade in the Chinese national standard, and the water absorption, apparent density, and bulk density were 3.54%, 1.19 g/cm^3^, and 0.81 g/cm^3^, respectively.(2)Due to the addition of magnesite tailings, cordierite (Mg_2_Al_4_Si_5_O_18_), magnesia–alumina spinel (MgAl_2_O_4_), and magnesium silicate (MgSiO_3_) were generated. This improved the compressive strength of the ceramsite. The uniform pore structure formed inside the ceramsite not only increased the compressive strength but also contributed to the solidification of heavy metals.(3)The concentration of heavy metals, namely, Zn, Mn, and Ni, in ceramsite decreased to 336, 222, and 125 mg/kg, respectively, less than one-third of the concentration in the raw pellets. All the heavy metals in the ceramsite existed in the form of FD at a level higher than 95%, enhancing stability. The leaching concentrations of Zn, Cu, Cr, and Ni were 0.722, 0.887, 0.201, and 0.249 mg/L, respectively (Pb was undetected), all far below the limit. In addition, the OPTI index of the ceramsite was only 236.92, less than one–fifth of the raw pellets, indicating that it is safe for the environment.

## Figures and Tables

**Figure 1 ijerph-19-11128-f001:**
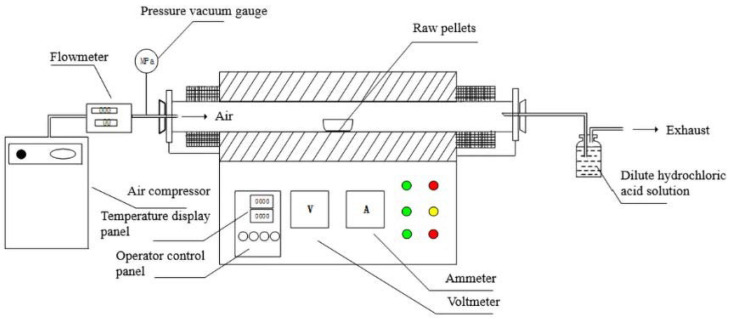
Diagram of experimental device.

**Figure 2 ijerph-19-11128-f002:**
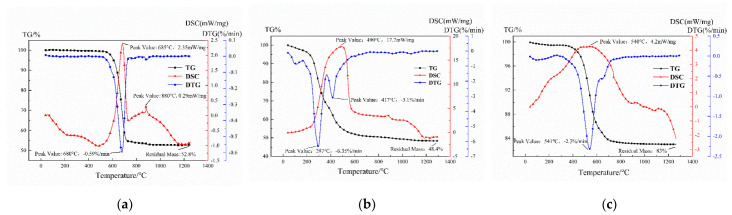
TG-DSC curves for (**a**) MTs, (**b**) SS, and (**c**) CG.

**Figure 3 ijerph-19-11128-f003:**
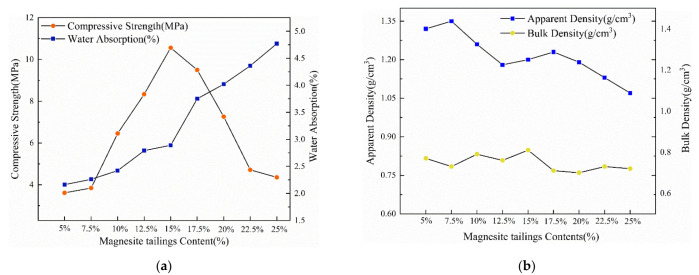
Properties of the ceramsite at different magnesite tailings content (**a**) compressive strength and water absorption, (**b**) apparent density and bulk density.

**Figure 4 ijerph-19-11128-f004:**
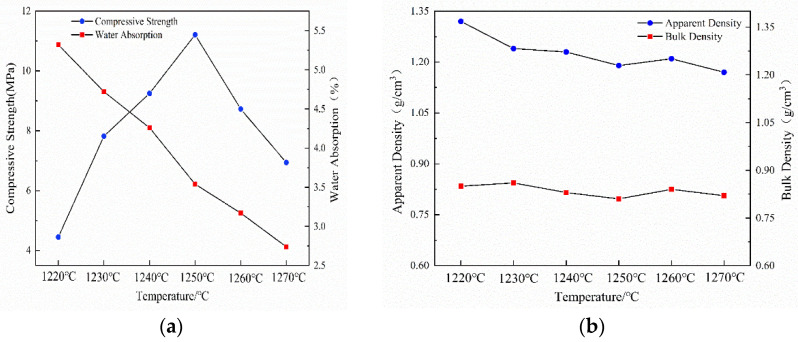
Properties of ceramsites at different temperatures (**a**) compressive strength and water absorption, and (**b**) apparent density and bulk density.

**Figure 5 ijerph-19-11128-f005:**
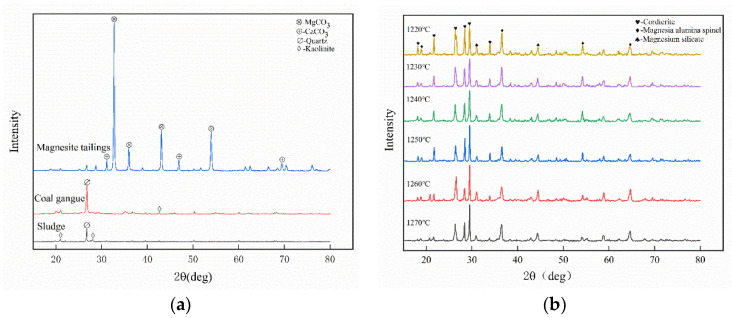
XRD patterns of raw materials and ceramsites at different sintering temperatures: (**a**) MTs, SS and CG; (**b**) ceramsites.

**Figure 6 ijerph-19-11128-f006:**
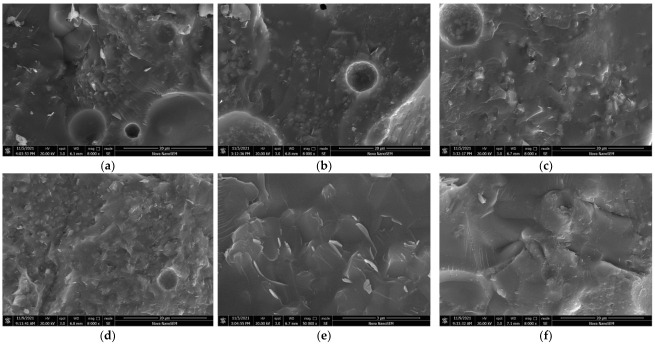
Surface microstructures of ceramsite at different sintering temperatures: (**a**) 1220 °C, (**b**) 1230 °C, (**c**) 1240 °C, (**d**) 1250 °C, (**e**) 1260 °C, (**f**) 1270 °C.

**Figure 7 ijerph-19-11128-f007:**
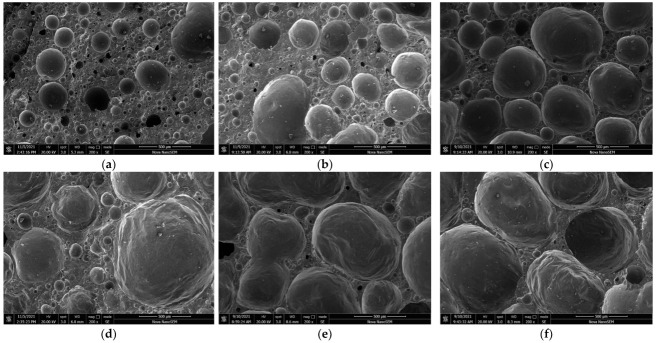
Interior microstructures of ceramsite at different sintering temperatures: (**a**) 1220 °C, (**b**) 1230 °C, (**c**) 1240 °C, (**d**) 1250 °C, (**e**) 1260 °C, (**f**) 1270 °C.

**Figure 8 ijerph-19-11128-f008:**
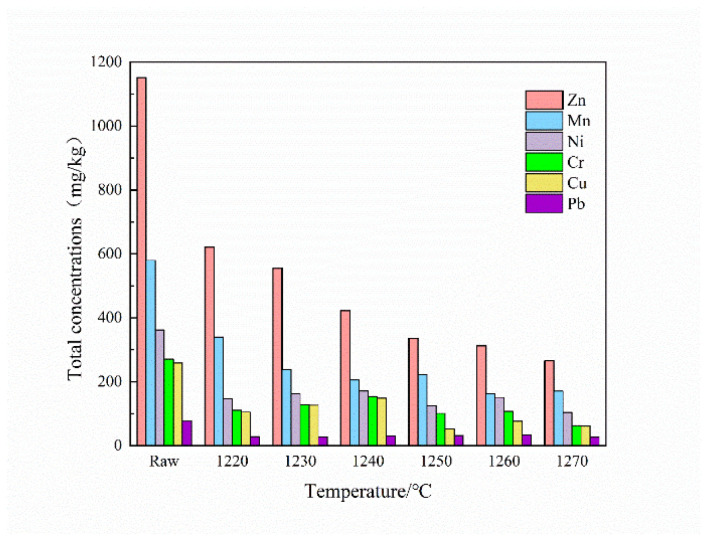
Total heavy metal concentrations at different sintering temperatures.

**Figure 9 ijerph-19-11128-f009:**
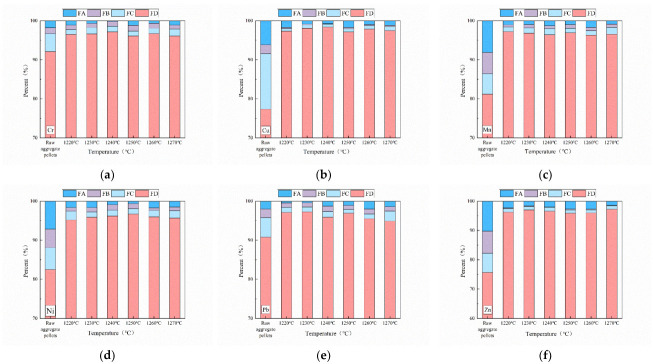
Morphological occurrence of heavy metals at different sintering temperatures: (**a**) 1220 °C, (**b**) 1230 °C, (**c**) 1240 °C, (**d**) 1250 °C, (**e**) 1260 °C, (**f**) 1270 °C.

**Figure 10 ijerph-19-11128-f010:**
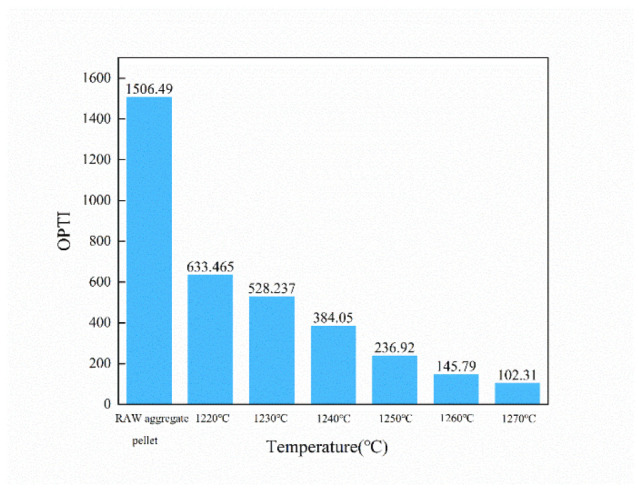
The OPTI of the ceramsites at different sintering temperatures.

**Table 1 ijerph-19-11128-t001:** Chemical composition of MTs, SS, and CG.

Composition (wt%)	MT	SS	CG
SiO_2_	10.15	39.14	66.79
Al_2_O_3_	3.693	15.75	30.35
MgO	33.59	-	-
Fe_2_O_3_	1.489	13.70	0.498
P_2_O_5_	-	12.04	-
CaO	2.91	13.91	0.175
Others	48.16	5.468	2.187

**Table 2 ijerph-19-11128-t002:** Proximate analysis and elemental analysis of MTs, SS, and CG.

Method		MT	SS	CG
Proximate analysis (wt%)	M_ad_	0.08 ± 0.01	1.06 ± 0.04	1.33 ± 0.01
	A_ad_	52.4 ± 0.01	56.88 ± 0.04	50.56 ± 0.06
	V_ad_	46.21 ± 0.06	36.92 ± 0.06	26.62 ± 0.01
	FC^a^_ad_	0.32	5.14	21.49
Elemental analysis (wt%)	C_ad_	13.247	20.28	30.98
	H_ad_	0.406	3.318	1.878
	N_ad_	-	1.21	0.8465
	S_ad_	-	0.478	1.505

**Table 3 ijerph-19-11128-t003:** The mixing ratios of SS, MTs, and CG.

Sample (wt%)	SS	CG	MTs
C1	55%	40%	5%
C2	52.5%	40%	7.5%
C3	50%	40%	10%
C4	47.5%	40%	12.5%
C5	45%%	40%	15%
C6	42.5	40%	17.5%
C7	40%	40%	20%
C8	37.5%	40%	22.5%
C9	35%	40%	25%

**Table 4 ijerph-19-11128-t004:** Leaching concentrations in the ceramsites (mg/L).

Sample	Raw	1220 °C	1230 °C	1240 °C	1250 °C	1260 °C	1270 °C	Limits
Zn	7.034	0.935	0.852	0.868	0.722	0.665	0.412	2.0
Ni	1.079	0.101	0.076	0.052	0.049	0.036	0.031	0.1
Cu	5.683	1.426	1.026	0.952	0.887	0.597	0.379	2.0
Cr	0.402	0.326	0.276	0.252	0.201	0.191	0.175	0.5
Pb	0.061	0.055	0.054	-	-	-	-	0.1

## Data Availability

Not applicable.

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
