# Peer review of "Study on the Properties and Heavy Metal Solidification Characteristics of Sintered Ceramsites Composed of Magnesite Tailings, Sewage Sludge, and Coal Gangue"

_ijerph, 2022, doi:10.3390/ijerph191711128_

Round 1

Reviewer 1 Report

The work of the authors « Study on the properties and heavy metal solidification characteristics of sintered ceramsite by magnesite tailings, sewage sludge and coal gangue» is written in a simple and accessible language for the reader. The research presented in the article is of high relevance. I believe that the authors are engaged in an important area for the disposal of industrial solid  waste. But despite the positive impression of the article, I have several recommendations and comments for the authors:

1.         The article provides a thermogravimetric analysis only for the feedstock (Fig. 2), but it seems to me that such an analysis should be done for all mixtures under study.

The analysis for plagiarism showed good results, the work is original.

The results of the study of the authors allow us to develop a direction for the beneficial processing of industrial waste. I think that the work can be published.

Author Response

  1. The article provides a thermogravimetric analysis only for the feedstock (Fig. 2), but it seems to me that such an analysis should be done for all mixtures under study.

Response: I appreciate your thoughtful suggestions. Unfortunately, due to limitations in time and the condition of the experiments, not every mixture was examined. Additionally, it wasn't previously thought that the mixture needed to undergo a thermogravimetric examination, but we will take this into account in the investigation that comes after this one.

Reviewer 2 Report

This is an important study. The manuscript is characterized by a high quality in all aspects. English should be checked by a professional language service or a native speaker.

I would suggest few amendments (respective lines):

14 has .......... the ratio

Edit this sentence, I suggest:

Line 14: The results showed that ceramsite had better properties when the following parametres were as displayed: ratio of SS: CG: MT 4.5:4:1.5; sintering temperature 1250 ℃; the compressive strength 11.2 Mpa (or it can be rounded to just 11; my major remark relates to significant figures, they should be up to 2-3 figures, according to measurement errors); water absorption 3.54%; and apparent and bulk densities 1.19 and 0.81 g/cm3, respectively. The strength was superior to more than twice the 900-density grade prescribed by the Chinese national standard.

19 what is FD?

19 The leaching concentrations of Zn and Ni from the ceramsite were 0.72 and 0.25 mg/L, respectively, lower than the prescribed regulatory limits.

20 what is OPTI index?

21 236.92 not, but either 240 or 237

22 The ceramsite, prepared from SS, CG and MT, exhibited not only excellent chemical (?) properties but it also proved to be environmentally safe material.

30 30%

81 was

88 was

98 and onwards: number of significant figures is too high, they should be corrected according to the relevant rules.

Author Response

  1. 14 has .......... the ratio

Edit this sentence, I suggest:Line 14: The results showed that ceramsite had better properties when the following parametres were as displayed: ratio of SS: CG: MT 4.5:4:1.5; sintering temperature 1250 ℃; the compressive strength 11.2 Mpa (or it can be rounded to just 11; my major remark relates to significant figures, they should be up to 2-3 figures, according to measurement errors); water absorption 3.54%; and apparent and bulk densities 1.19 and 0.81 g/cm3, respectively. The strength was superior to more than twice the 900-density grade prescribed by the Chinese national standard.

Response: Thank you very much for your positive comment and pointing out our negligence. We have corrected this error in the manuscript.

  1. 19 what is FD?

Response: Thank you very much for your positive comment and pointing out our negligence. We have added an explanation of residue state (FD) to the manuscript.

  1. 19 The leaching concentrations of Zn and Ni from the ceramsite were 0.72 and 0.25 mg/L, respectively, lower than the prescribed regulatory limits.

Response: Thank you very much for your positive comment and pointing out our negligence. We have corrected this error in the manuscript.

  1. 20 what is OPTI index?

Response: Thank you very much for your positive comment and pointing out our negligence. We have added an explanation of overall pollution toxicity index (OPTI) to the manuscript.

  1. 21 236.92 not, but either 240 or 237

Response: Thank you very much for your positive comment and pointing out our negligence. We have corrected this error in the manuscript.

  1. 22 The ceramsite, prepared from SS, CG and MT, exhibited not only excellent chemical (?) properties but it also proved to be environmentally safe material.

Response: Thank you very much for your positive comment and pointing out our negligence. We changed 236.92 to 240.

  1. 30 30%

Response: Thank you very much for your positive comment and pointing out our negligence. We changed 28.85% to 30%.

  1. 81 was

Response: Thank you very much for your positive comment and pointing out our negligence. We have revised the original into “The purpose of this study was to determine the feasibility of using MTs, SS and CG to prepare environmentally friendly artificial ceramsite with excellent performance, this provides a solution for the resource utilization of industrial solid waste.”

  1. 88 was

Response: Thank you very much for your positive comment and pointing out our negligence. We have revised the original into “The moisture content of SS was high (about 82%).”

  1. 98 and onwards: number of significant figures is too high, they should be corrected according to the relevant rules.

Response: Thank you very much for your positive comment and pointing out our negligence. We have corrected this error in the manuscript.

Table 1 Chemical composition of MT, SS and CG.

Composition(wt%)

MT

SS

CG

SiO2

10.15

39.14

66.79

Al2O3

3.693

15.75

30.35

MgO

33.59

-

-

Fe2O3

1.489

13.70

0.498

P2O5

-

12.04

-

CaO

2.91

13.91

0.175

Others

48.16

5.468

2.187

Reviewer 3 Report

1. Superscripts should be used for some units (line 62). 

2. Abbreviations such as MT, SS and CG should be explained in the text (not everybody will read the abstract).

3. ,, The details of OPTI were shown in our previous study (A new inte- 135 grated evaluation method of heavy metals pollution control during melting and sintering 136 of MSWI fly ash)."- reference should be given. 

4. ,,Where: F is the pressure (KN), s is the cross-sectional area under pressure(mm2 ), and Fa 140 is, compressive strength (MPa)."- from what is written it can be assumed that F is force not pressure. Please describe equation more clearly. 

5. ,,Where: m0 is the quality of ceramsite (g), m1 is the quality of submerged ceramsite (g), and 143 w is water absorption rate (%)." - rom what is written it can be assumed that m is water content not quality. Please describe equation more clearly. 

6. Line 146 - it should be M not m.

7. Line 154-155 - what does loss weightlessness mean? Please define or use commonly known terms. 

8. Line 157 - please correct the sentence it is not grammatical.

9. Line 160 - please correct the sentence it is not grammatical.

10. Language quality and clarity of results presentation should be corrected in the whole article. 

11. Line 189 - it would be better to give the values he compressive strength alongside reference to the standard.

12. ,,High-temperature sintering of ceramsite can effectively reduce the leaching concentrations" - but with rising temperature whe metals will be evaporated and the process gas will require further treatment for removal of haevy metals. Is it economically justified?

Author Response

We have commissioned MDPI English Editing to revise and improve the whole manuscript.

  1. Superscripts should be used for some units (line 62).

Response: Thank you very much for your positive comment and pointing out our negligence. We have revised the original into “Bouachera et al. [14] mixed sewage sludge, waste glass and clay at a certain quality ratio (3:3:4) to reduce the sintering temperature and to lower the bulk density of ceramsite to 0.84g/cm3, but the compressive strength was only 2.51 MPa.”

  1. Abbreviations such as MT, SS and CG should be explained in the text (not everybody will read the abstract).

Response: Thank you very much for your positive comment and pointing out our negligence. We have added an explanation of sewage sludge(SS), magnesite tailings(MTs), and coal gangue(CG) to the manuscript.

  1. “The details of OPTI were shown in our previous study (A new integrated evaluation method of heavy metals pollution control during melting and sintering of MSWI fly ash)."- reference should be given.

Response: Thank you very much for your positive comment and pointing out our negligence. We have cited references in the original article.

  1. “Where: F is the pressure (KN), s is the cross-sectional area under pressure (mm2), and Fa is, compressive strength (MPa).” - from what is written it can be assumed that F is force not pressure. Please describe equation more clearly.

Response: Thank you very much for your positive comment and pointing out our negligence. We have revised the original into “where F is the force (KN), s is the cross-sectional area under pressure (mm2), and Fa is the compressive strength (MPa).”

  1. “Where: m0 is the quality of ceramsite (g), m1 is the quality of submerged ceramsite (g), and w is water absorption rate (%).” - rom what is written it can be assumed that m is water content not quality. Please describe equation more clearly.

Response: Thank you very much for your positive comment and pointing out our negligence. We have revised the original into “where m0 is the quality of the ceramsite (g), m1 is the water content of the submerged ceramsite (g), and  is the water absorption rate (%).”

  1. Line 146 - it should be M not m.

Response: Thank you very much for your positive comment and pointing out our negligence. We have revised the original into “where M is the weight of the ceramsite (g), Vt is the volume of the ceramsite (cm3), and  is the apparent density (g/cm3).”

  1. Line 154-155 - what does loss weightlessness mean? Please define or use commonly known terms.

Response: Thank you very much for your positive comment and pointing out our negligence. We have revised the original into “From the TG-DSC curve of the magnesite tailings (Figure 2a), it can be observed that the weightlessness is 47% between 580 ℃ and 700 ℃, and the maximum weight loss rate is 0.59%/min at 680 ℃.”

  1. Line 157 - please correct the sentence it is not grammatical.

Response: Thank you very much for your positive comment and pointing out our negligence. We have revised the original into “This is mainly due to the thermal decomposition of MgCO3 [24,25]. The maximum heat flux at 685℃ is also confirmed by the differential scanning calorimeter curve to be 2.35 mW/mg.”

  1. Line 160 - please correct the sentence it is not grammatical.

Response: Thank you very much for your positive comment and pointing out our negligence. We have revised the original into “The first stage of weightlessness is about 18%, the start time of weightlessness is 240min, the end time of weightlessness is 300min, the maximum weightlessness rate is 6.35%/℃, and the weightlessness time is 297min.”

  1. Language quality and clarity of results presentation should be corrected in the whole article.

Response: Thank you for your kind suggestions. We've turned it over to a professional body for polishing

  1. Line 189 - it would be better to give the values he compressive strength alongside reference to the standard.

Response: Thank you very much for your positive comment and pointing out our negligence. We have revised the original into “The compressive strength of the ceramsite (SS: CG: MT=4.5:4:1.5) reaching 7.85 MPa, higher than that of the others, and the compressive strength is superior to the 900 density grade (5Mp) in the Chinese national standard GB / T 17431.1 – 2010 [34].”

  1. “High-temperature sintering of ceramsite can effectively reduce the leaching concentrations" - but with rising temperature whe metals will be evaporated and the process gas will require further treatment for removal of haevy Is it economically justified?

Response: Thank you very much for pointing out our negligence. We did not consider the economic benefits of this part before, and we will include this part in the subsequent research.

Reviewer 4 Report

The authors used magnesite tailings, sewage sludge and coal gangue to produce ceramsite, and further studied the characteristics. Overall, this is an interesting study. However, the Language does not meet the standard, which needs to be improved substantially. Below I just list some examples. In my opinion, the English in the whole manuscript needs to be edited. I’m sorry I can’t recommend its acceptance.

L13: grammar error, “influence…were”, correct “were” as “was”

L14: correct “have” as “has”, because “ceramsite”

L15: revise “=” as “was”; “whose” is not clear;

L17: grammar error: “.,”

L19: Spell out “FD” at the first appearance

L20: what is “the limits”?

L20: What does “OPTI” mean? Spell out it.

L21: it is difficult to understand “less more than” use “less” or “more”

L22: grammar error, “were not only have”

L10-22: You can use past tense or present tense, but can’t mix them.

L31: the comma is wrong.

L32: What does “it” represent? Note, “tailings” is plural.

L34-35: This is not a complete sentence, with wrong grammar.

L57: Nakouzi et al.

L58: remove ”Gong”

L60: correct “Rachida bouachera” as “Bouachera

L69: wrong sentence.

Table 1, Table 3: The proportion is based on weight or volume?

L178: where is the “3.2”?

Author Response

We have commissioned MDPI English Editing to revise and improve the whole manuscript.

  1. L13: grammar error, “influence…were”, correct “were” as “was”

Response: Thank you very much for your positive comment and pointing out our negligence. We have revised the original into “The influence of the material ratio and sintering temperature on the properties of the ceramsite was investigated.”

  1. L14: correct “have” as “has”, because “ceramsite”

Response: Thank you very much for your positive comment and pointing out our negligence. We have revised the original into “The results show that the ceramsite had better properties when the following parameters were used: a ratio of SS: CG: MT of 4.5:4:1.5; a sintering temperature of 1250 ℃; a compressive strength of 11.2 Mpa (or it can be rounded to 11; our major remark relates to significant figures they should be up to 2-3 figures, according to measurement errors); a water absorption of 3.54%; and apparent and bulk densities of 1.19 and 0.81 g/cm3, respectively.”

  1. L15: revise “=” as “was”; “whose” is not clear;

Response: Thank you very much for your positive comment and pointing out our negligence. We have revised the original into “The results show that the ceramsite had better properties when the following parameters were used: a ratio of SS: CG: MT of 4.5:4:1.5; a sintering temperature of 1250 ℃; a compressive strength of 11.2 Mpa (or it can be rounded to 11; our major remark relates to significant figures they should be up to 2-3 figures, according to measurement errors); a water absorption of 3.54%; and apparent and bulk densities of 1.19 and 0.81 g/cm3, respectively.”

  1. L17: grammar error: “.,”

Response: Thank you very much for your positive comment and pointing out our negligence. We have corrected this error in the manuscript.

  1. L19: Spell out “FD” at the first appearance

Response: Thank you very much for your positive comment and pointing out our negligence. We have added an explanation of residue state (FD) to the manuscript.

  1. L20: what is “the limits”?

Response: Thank you very much for your positive comment and pointing out our negligence. We have added an explanation of the limits ( 2.0 and 0.1 mg/L ) to the manuscript.

  1. L20: What does “OPTI” mean? Spell out it.

Response: Thank you very much for your positive comment and pointing out our negligence. We have added an explanation of overall pollution toxicity index (OPTI) to the manuscript.

  1. L21: it is difficult to understand “less more than” use “less” or “more”

Response: Thank you for your kind advice. We have revised the original into “The overall pollution toxicity index (OPTI) was only 240, less than that of raw pellets, in-dicating that the environmental risk is low.”

  1. L22: grammar error, “were not only have”

Response: Thank you for your kind advice. We have revised the original into “Not only did the ceramsite, prepared from SS, CG and MT, exhibit excellent chemical properties, but it also proved to be an environmentally safe material.”

  1. L10-22: You can use past tense or present tense, but can’t mix them

Response: Thank you for your kind advice. We have corrected the misnomer here.

  1. L31: the comma is

Response: Thank you for your kind advice. We have revised the original into “China has large reserves and a high mining volume of magnesite, the proved reserves are about 3.6 billion tons, accounting for 30% of the world reserves, and the annual mining volume reach 20 million tons.”

  1. L32: What does “it” represent? Note, “tailings” is plural.

Response: Thank you for your kind advice. We have revised the original into “If the tailings generated by mining are not properly treated, they will pollute the environ-ment and endanger human life [2].”

  1. L34-35: This is not a complete sentence, with wrong grammar.

Response: Thank you very much for your positive comment and pointing out our negligence. We have revised the original into “The preparation methods of MgCO3 and MgO comprise calcination, hydration, carbonization and thermal decomposition methods, and the preparation method of magnesiaalumina spinel cementitious material is sintering [3].”

  1. L57: Nakouzi et al.

Response: Thank you very much for your positive comment and pointing out our negligence. We have revised the original into “Nakouzi et al [12] first prepared ceramsite by using sludge in 1998, and the ceramsite formed a hard surface during high-temperature sintering.”

  1. L58: remove ”Gong”

Response: Thank you very much for your positive comment and pointing out our negligence. We have revised the original into “Cheng et al. [13] used sludge, fly ash and oyster shell to sinter ceramsite, and they obtained an optimal phosphate adsorption capacity of 4.51/mg at a sintering temperature of 1050℃.”

  1. L60: correct “Rachida bouachera” as “Bouachera”

Response: Thank you for your kind advice. We have corrected the misnomer here “Bouachera et al. [14] mixed sewage sludge, waste glass and clay at a certain quality ratio (3:3:4) to reduce the sintering temperature and to lower the bulk density of ceramsite to 0.84g/cm3, but the compressive strength was only 2.51 MPa.”

  1. L69: wrong sentence.

Response: Thank you very much for your positive comment and pointing out our negligence. We have revised the original into “At present, there are few studies on the preparation of ceramsite with magnesite tailings as raw materials.”

  1. Table 1, Table 3: The proportion is based on weight or volume?

Response: Thank you very much for your positive comment and pointing out our negligence. The proportion is based on weight. And we have added identifiers to the manuscript.

  1. L178: where is the “3.2”?

Response: Thank you very much for your positive comment and pointing out our negligence. We have corrected it in the manuscript. “3.2 The properties of sintered ceramsite”

Round 2

Reviewer 4 Report

The current version is acceptable.